# Evaluation of Serum Zinc Status of Pregnant Women in the China Adult Chronic Disease and Nutrition Surveillance (CACDNS) 2015

**DOI:** 10.3390/nu13041375

**Published:** 2021-04-20

**Authors:** Xiao-Bing Liu, Jia-Xi Lu, Li-Juan Wang, Yi-Chun Hu, Rui Wang, De-Qian Mao, Jian Huang, Li-Yun Zhao, Xiao-Guang Yang, Li-Chen Yang

**Affiliations:** Key Laboratory of Trace Element Nutrition of National Health Commission, National Institute of Nutrition and Health of Chinese Center for Disease Control and Prevention, Beijing 100050, China; liuxb@ninh.chinacdc.cn (X.-B.L.); lujx@ninh.chinacdc.cn (J.-X.L.); wanglj@ninh.chinacdc.cn (L.-J.W.); huyc@ninh.chinacdc.cn (Y.-C.H.); wangrui@ninh.chinacdc.cn (R.W.); maodq@ninh.chinacdc.cn (D.-Q.M.); huangjian@ninh.chinacdc.cn (J.H.); zhaoly@ninh.chinacdc.cn (L.-Y.Z.); xgyangcdc@163.com (X.-G.Y.)

**Keywords:** serum zinc, pregnant women, zinc deficiency, China Adult Chronic Disease and Nutrition Surveillance

## Abstract

**Objective:** The purpose of this study was to evaluate serum zinc status of pregnant women in the China Adult Chronic Disease and Nutrition Surveillance (CACDNS) in 2015–2016. **Methods:** A total of 7147 apparently healthy pregnant women were randomly selected in 302 national monitoring sites. Information on age, race, residence region, education, pregnancy, and family income per annum was collected, and the concentration of serum zinc was determined. The evaluation of serum zinc status was further performed according to the recommendations by the International Zinc Nutrition Consultative Group (IZiNCG). **Results:** The median concentration of serum zinc was 858.9 μg/L with an interquartile range (IQR) of 712.9 μg/L and 1048.9 μg/L, while the overall prevalence of zinc deficiency was 3.5% with a 95% confidence interval (CI) of 3.0% and 3.9%. Serum zinc status of pregnant women changed greatly in the different categories, particular in pregnancy and family income per annum (*p* < 0.05), but no significant difference was observed in the prevalence of zinc deficiency (*p* > 0.05). **Conclusions:** The lower prevalence of zinc deficiency generally indicated a better zinc status for pregnant women in the CACDNS in 2015–2016. However, a well-designed evaluation system of zinc status for pregnant women should be continually optimized and improved by inducing more parameters such as biochemical, dietary, or functional indicators.

## 1. Introduction

Zinc is generally considered as the most important trace element that functions as a catalyst, structural element and regulatory role in numerous metabolic processes, including DNA transcription and gene expression, signal transduction, and endocrine function [1]. To date, it was reported that zinc is mainly involved in over 300 enzymes and more than 2000 transcription factors. Zinc deficiency can cause multiple dysfunctions and health hazards including stunted growth, cognitive impairment, poor pregnancy outcomes, and increased infections [2]. Currently, to our knowledge, at least the 17% of global population is at risk of inadequate zinc intake, making zinc deficiency one of the most prevalent micronutrient deficiencies worldwide, and affecting children, pregnant and lactating women, and elderly people [3].

Pregnancy is a special period with much more nutritional demands, and pregnant women will undergo obvious anatomical and physiological changes [4]. If zinc intake is inadequate before pregnancy, zinc deficiency could be exacerbated due to increased various requirements for hematopoiesis, growth, and fetal development, further resulting in a series of more adverse birth outcomes including intrauterine growth retardation, preterm delivery, low birth weight, and increased morbidity and mortality of the fetus [5,6]. In addition, zinc participates in energy metabolism for carbohydrates, proteins, and lipids [7]. The potential association of zinc status and increased gestational age is receiving more attention in pregnant women. However, albeit of well-known multiple detrimental consequences for zinc deficiency, there still is a paucity of such data to describe zinc nutritional status in China, so far. Indeed, it is firstly important to evaluate zinc status accurately and to further differentiate between normal and abnormal individuals exactly. Regrettably, the current dilemma to be faced is how to examine zinc status accurately to improve zinc nutrition for Chinese pregnant women.

The evaluation of zinc nutritional status in individuals always is a great challenge for methodology due to enormous expenditure, although currently several biomarkers are widely suggested, and involving the determination of zinc content in blood, serum, erythrocytes, hair, and zinc transporters. Of these, serum zinc is one of the most widely used and accepted indicators reflecting zinc status in large populations despite its relative poor sensitivity and its imperfect specificity [8]. More importantly, as a traditional indicator, serum zinc is jointly recommended by the World Health Organization (WHO), United Nations International Children’s Emergency Fund (UNICEF), International Atomic Energy Agency (IAEA), and International Zinc Nutrition Consultative Group (IZiNCG) [9]. Therefore, the purpose of this study is to obtain the reliable data of serum zinc of pregnant women in the nationally representative samples derived from the China Adult Chronic Disease and Nutrition Surveillance (CACDNS) in 2015–2016 and further to delineate zinc status distinctly so as to focus on the precise nutrition policies.

## 2. Materials and Methods

### 2.1. Study Population

This present study is a nationally population-based sample of Chinese pregnant women derived from the CACDNS in 2015–2016, including the 302 monitoring sites widely distributed in 31 provinces, autonomous regions, and municipalities directly under the central government of China (except Taiwan, Hong Kong, and Macau). A multistage stratified cluster random sampling method was employed in combined with probability proportionate to population size sampling, and the sampling schemes were detailly reported in previous literature [10]. The protocol was drawn in accordance with the guidelines of the Declaration of Helsinki [11] and were approved by the Ethical Committee of the Chinese Center for Disease Control and Prevention (Grant No. 201519-A). Information on age, race, residence region, education, family income per annum, and anthropometric indicators was together collected, and written informed consents were provided by each subject after the nature of the CACDNS in 2015–2016 explained. Because the selection of pregnant women was generally set at 30 at each monitoring site, in the current study, the inclusion criteria of the study population were further prescribed according to pregnancy (Trimester 1, 2, and 3) and residence region (urban, rural). Questionnaire information was required to be completed as much as possible, and the serum specimens were required to be: no missing, no hemolysis, adequate sample capacity, and well-preserved status.

### 2.2. Serum Sample Collection and Laboratory Analyses

The peripheral blood of 5 mL was severally collected by venipuncture from each participant after over 8 h of fasting in a sterile vacutainer tube. Blood samples were allowed to stand for 30 min at room temperature, and the coagulate was centrifuged in a disposable sterile vacuum plastic tubes at 3000 rpm for 10 min, and serum specimen was then separated, repackaged into several tubes, and stored in clean, metal-free, polypropylene tubes at −20 °C for a short term. Subsequently, serum samples were shipped on dry ice to the Laboratory of National Institute of Nutrition and Health of Chinese Center for Disease Control and Prevention and stored in a deep freeze at −80 °C until analysis. In this study, serum samples were chosen according to the sampling scheme and freeze-thawed at room temperature. C-reactive protein (CRP) and alpha-1-acid glycoprotein (AGP) were determined by the automatic blood biochemical analyzer Cobas Integra 400 plus analyzer (Roche Diagnostic, Rotkreuz, Switzerland). The serum samples to be analyzed were then diluted (1:20) using the diluent of 0.5% (*v*/*v*) high pure nitric acid solution. Serum zinc concentration was determined using inductively coupled couple mass spectrometer (ICP–MS). Commercially available quality control samples (Sero norm™, Level-1, Level-2, Olso, Norway) were employed to monitor precision and accuracy of the analyses at an interval of 10 samples during the period of whole analysis. The recoveries ranged from 90.0 to 95.3%, and the inter-day precisions were 2.0%, and intra-day precisions were 2.3–8.7%.

### 2.3. Statistical Analyses

All statistical analyses were conducted using SAS 9.2 (SAS Institute, Cary, NC, USA). In the process of data cleaning, the data of serum zinc were required to be removed when relevant questionnaire variables were missing or the inflammatory factor such as C-reactive protein (CRP) > 5 mg/g or alpha-1-acid glycoprotein (AGP) > 1 g/L. The data normality was examined by the Shapiro-Wilk test, and the outliers were removed by the Tukey’s method. The concentration of serum zinc was finally expressed as median and interquartile ranges (IQR) for abnormal distribution. The cut-off value of zinc deficiency in pregnant women was defined as serum zinc concentration <560 μg/L in the first trimester and <500 μg/L in the second and third trimester [12]. The Mann-Whitney U test was used to examine the differences in serum zinc concentration in age group, race, residence region, educational level, pregnancy, and family income per annum. The Pearson’s chi–square hypothesis test was used followed by Fisher’s Exact Test (since two cells were under five) comparing the prevalence of zinc deficiency among the different subgroups. Finally, *p* < 0.05 was considered statistically significant.

## 3. Results

### 3.1. Serum Zinc Status of Pregnant Women

In this study, a total of 7147 pregnant women were originally selected in the 302 monitoring sites, while some observations were finally removed for incomplete information collection during the data cleaning. Age was 28.1 ± 5.3 years, BMI was 24.4 ± 4.0 kg/m^2^, and family income per annum equated to approximately 34.5 ± 9.9 thousands Yuan. The serum zinc status of pregnant women in the CACDNS in 2015–2016 are shown in Table 1. The median concentration of serum zinc was 858.9 μg/L with an IQR of 712.9 μg/L to 1048.9 μg/L. We observed that serum zinc concentration varied slightly, associating with increased age, alteration of race and residence region, increased educational levels, and gestational age, as well as with family income per annum. Of these, serum zinc status significantly decreased with increased gestational age but positively elevated and accompanied with increased family income per annum (*p* < 0.05).

### 3.2. Prevalence of Zinc Deficiency and Multivariate Logistic Regression Analysis

As shown in Table 2, the overall prevalence of zinc deficiency was 3.5% with a 95% CI of 3.0 to 3.9%. The risk of zinc deficiency was kept at low-level status and there was only a trivial fluctuation among the different categories such as age group, race, residence region, gestational age, and family income per annum (*p* > 0.05), in addition to educational level (*p* < 0.05), indicating that higher educational level was associated with a lower risk of zinc deficiency. Moreover, the association of zinc deficiency was employed to explore the risk factors further by multiple logistic regression analysis. However, from these data, it seems that there was no significant risk factor affecting zinc deficiency.

## 4. Discussion

This current study obtained the reliable data of serum zinc of pregnant women based on a nationally representative sample in the CACDNS in 2015–2016. The overall concentration of serum zinc was 858.9 μg/L, while the prevalence of zinc deficiency was 3.5%. Serum zinc concentration was positively associated with age group, educational level, and family income per annum, and negatively decreased with advanced gestational age, but no similar changes were observed in the prevalence of zinc deficiency among the different categories. The lower prevalence of zinc deficiency further indicated a lower risk of zinc deficiency of pregnant women in the CACDNS in 2015–2016, which is far below the cutoff value of 20% suggested by the IZiNCG; however, it still remains a continuing concern for nutrition and health-monitoring in Chinese pregnant women.

At present, serum zinc is endorsed as the best available biomarker in the absence of a gold standard. Regrettably, the assessment of zinc status could still be considered particularly puzzling and should be considered in pregnancy for accumulative physiologic adjustments in zinc metabolism. Normally, serum zinc will decrease, associating with the expansion of maternal blood volume and reduced serum albumin concentration, and increased concentrations of circulating estrogen [13]. In our study, significant differences were indeed observed in serum zinc level, with a distinct decline with increased gestational age (*p* < 0.0001) and a trivial rising with increased family income per annum (*p* < 0.05). As for the change, we considered that it could partially attributed to the requirements of the fetus to reduce serum zinc level further in Chinese pregnant women. There were no significant differences in the other different categories (*p* > 0.05). By contrast, more economic development generally indicates a better zinc status due to the increased chance of dietary zinc consumption in pregnant women, as well as improvement in educational level. Notably, there was a contrasting difference in the prevalence of zinc deficiency in pregnancy. As for these disparities, they may be due to insufficient dietary zinc intake in the first trimester, and awareness is relatively weak in the early stage of pregnancy, while more various nutrient intake will increase stepwise later.

As a rule, serum zinc level could fall or rise in response to some complicated factors unrelated to zinc status or dietary zinc intake, such as infection, inflammation, exercise, stress or trauma, circadian variation, and fasting status [14]. Further, there were an inevitable limitation that only a smaller proportion of the body zinc is presenting in the serum pool in the body [15]. Importantly, it was easily contaminated by exterior exposure and inappropriate handling during the collection of biological specimens. Moreover, diurnal variation in circulating zinc probably is an indirect reflection of metabolic changes, because some changes may occur as a result of normal circadian variation in metabolism [16]. Besides, in sample selection, time of day, food and the consumed meals’ interval, recent exercise, or other forms of stress all can result in obvious fluctuations [17]. In our study, serum zinc concentration was obviously higher than the value reported in previous Chinese studies [18], similar to the data in some countries and regional documents, but slightly lower than the results of USA [19], EU [20,21], Korea [22], and Japan [23]. In this regard, there was a trivial difference in zinc deficiency prevalence as compared to the significant changes in serum zinc concentration. We further considered that these differences could be attributed to various dietary patterns, different laboratory methods, and subjects’ characteristics; all of these can result in a broad variability in serum zinc status.

There are some strengths and limitations in our study. First, the current study is a relatively reliable evaluation of serum zinc status in a nationally representative sample of Chinese pregnant women in the CACDNS in 2015–2016. However, it should still be prudent enough for partial missing data when extrapolating to other populations. Second, serum zinc is a reliable biomarker in assessing zinc status in the population, thus supporting our conclusion. Yet, there is only the serum zinc data and a lack of dietary zinc intake for Chinese pregnant women, making it very difficult for further analysis. Thus, a well-designed evaluation of zinc status of pregnant women is urgently required to be optimized.

## 5. Conclusions

Only nearly 3.5% of zinc deficiency generally means a lower public health risk, but it still requires constant focus in pregnant women. Additionally, we further suggest that the systematical evaluation of zinc status should be improved by introducing more sensitive and specific biochemical indicators, and dietary and functional parameters.

## Figures and Tables

**Table 1 nutrients-13-01375-t001:** Serum zinc status of pregnant women in the CACDNS in 2015–2016.

Variables	N (Percent %)	SZC (μg/L)		*p* Value
		Median	IQR	
All	7147	858.9	712.9–1048.9	–
Age group (Years)				0.06
18–25	2393 (34.8)	849.7	712.9–1029.1	
26–30	2930 (42.6)	860.4	713.2–1053.4	
31–44	1552 (22.6)	868.7	711.2–1074.2	
Race				0.06
Han	5971 (86.9)	855.7	710.2–1044.2	
Minority	904 (13.1)	870.3	722.6–1076.7	
Residence region				0.64
Urban	4185 (61.5)	856.2	709.1–1054.2	
Rural	2623 (38.5)	860.8	716.9–1040.5	
Educational level				0.18
Primary school and below	483 (7.0)	855.0	712.4–1023.1	
Middle school	3509 (50.8)	857.2	706.7–1043.9	
College and above	2913 (42.2)	860.1	720.6–1057.7	
Pregnancy				<0.0001
First trimester	1995 (29.0)	924.3 ^a^	764.6–1119.4	
Second trimester	2586 (37.6)	845.8 ^b^	705.5–1030.2	
Third trimester	2294 (33.4)	822.2 ^c^	679.2–997.2	
Family income per annum (Yuan)				0.014
<10,000	558 (12.6)	833.5 ^a^	710.2–1015.1	
10,000–20,000	2186 (49.2)	862.2 ^b^	720.6–1050.3	
20,000–30,000	1272 (28.6)	855.5 ^b^	717.2–1051.0	
>30,000	428 (9.6)	909.9 ^b^	725.9–1090.5	

Note: CACDNS: China Adult Chronic Disease and Nutrition Surveillance N, number; SZC, serum zinc concentration; IQR, range interquartile; Values not sharing the same superscript letter (a–c) denote a significant difference among subgroups, *p* < 0.05.

**Table 2 nutrients-13-01375-t002:** Prevalence of zinc deficiency and multivariate logistic regression analysis in Chinese pregnant women.

Variables	Zinc Deficiency (%)			OR	95% CI	*p* Value
	Prevalence	95% CI	*p* Value			
Total population	3.5	3.0–3.9	–	–	–	–
Age group (Years)			0.63	–	–	–
18–25	3.9	3.2–4.7		–	–	–
26–35	3.2	2.6–3.9		0.91	0.68–1.24	0.56
36–44	3.9	3.0–5.0		1.09	0.78–1.53	0.61
Race			0.54			
Han	3.7	3.2–4.2		–	–	–
Minority	3.2	2.2–4.6		0.98	0.65–1.46	0.90
Residence region			0.18			
Urban	4.0	3.4–4.6		–	–	–
Rural	2.9	2.3–3.7		0.77	0.58–1.02	0.07
Educational level			0.03			
Primary school and below	5.5	3.6–8.0		–	–	–
Middle school	3.5	2.9–4.2		1.08	0.68–1.70	0.75
College and above	3.4	2.8–4.1		1.00	0.62–1.62	0.99
Pregnancy			0.18			
First trimester	4.3	3.4–5.2		–	–	–
Second trimester	2.8	2.2–3.5		0.73	0.53–1.02	0.06
Third trimester	3.9	3.2–4.8		1.04	0.77–1.42	0.78
Family income per annum (Yuan)			0.78			
<10,000	2.9	1.7–4.6		–	–	–
10,000–30000	3.5	2.8–4.3		1.07	0.79–1.45	0.67
30,000–50,000	3.7	2.7–4.9		1.12	0.78–1.60	0.55
>50,000	3.5	2.0–5.7		1.05	0.60–1.84	0.87

Note: OR, odds ratio; CI, confidence interval.

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
