# Peer review of "Evaluation of Serum Zinc Status of Pregnant Women in the China Adult Chronic Disease and Nutrition Surveillance (CACDNS) 2015"

_nutrients, 2021, doi:10.3390/nu13041375_

Round 1

Reviewer 1 Report

Reviewer’s Comments:

The manuscript "Evaluation of serum zinc status of pregnant women in the Chinese Nutrition and Health Surveillance (CNHS) 2015−2016" by Liu et al presents the results of a study that was conducted to evaluate the serum zinc status of pregnant women in the Chinese Nutrition and Health Surveillance (CNHS) 2015−2016.

Comments:

  1. Please include the exclusion and inclusion criteria for the study population in a table.
  2. Can the results of this study that examined a limited number of subjects, be extrapolated to other populations?
  3. Please add a brief paragraph on “future directions to this study” at the end of the discussion/conclusions section.
  4. Please be consistent with the style of references: some of the references have a ‘DOI’ and page numbers while others don’t.
  5. Please proofread for spelling and grammatical errors.

Author Response

Point 1: Please include the exclusion and inclusion criteria for the study population in a table.

Response 1: Many thanks for your suggestions. The exclusion and inclusion criteria have been further stated in the revised manuscript [Line 101-107].

Point 2: Can the results of this study that examined a limited number of subjects, be extrapolated to other populations?

Response 2: Obviously not. The current study had evaluated the serum zinc status of Chinese pregnant women in the CNHS 2015-2016. However, there still is a lack of some important information such as partial missing data, dietary intake and other biochemical parameters, making the results are more difficult to extrapolate to other populations.

Point 3: Please add a brief paragraph on “future directions to this study” at the end of the discussion/conclusions section.

Response 3: Thanks a lot for your valuable advice. The future direction to this study had been added in the conclusion section of the revised manuscript [Line 254-256].

Point 4: Please be consistent with the style of references: some of the references have a ‘DOI’ and page numbers while others don’t.

Response 4: Thanks for your kindly reminder. The references had been proofread again according to the submission requirement.

Point 5: Please proofread for spelling and grammatical errors.

Response 5: Thanks. The spelling and grammatical errors have been corrected in the revised manuscript.

Reviewer 2 Report

The manuscript presents an interesting subject. Generally, it is well designed and described. I have only a few issues:

  1. In my opinion, the including and excluding criteria should be included in the ‘materials and method’ part.
  2. In limitation Authors should added information about the lack of the dietary intake parameters.
  3. Conclusions should be rewritten. It is a summary, not a real conclusion.

Author Response

Point 1: In my opinion, the including and excluding criteria should be included in the ‘materials and method’ part.

Response 1: We all agree with your suggestion. The excluding and including criteria have been added in the revised manuscript. [Line 101-107]

Point 2: In limitation Authors should added information about the lack of the dietary intake parameters.

Response 2: The limitation had been pointed out and included in our revised manuscript. In addition, we further suggest that dietary zinc intake and other parameters should be listed as the main future directions in assessing zinc status.

Point 3: Conclusions should be rewritten. It is a summary, not a real conclusion.

Response 3: Thank you very much for your pertinent suggestion. The conclusion had been rewritten in the revised manuscript.